# Ovarian-Cancer-Associated Extracellular Vesicles: Microenvironmental Regulation and Potential Clinical Applications

**DOI:** 10.3390/cells10092272

**Published:** 2021-09-01

**Authors:** Priyakshi Kalita-de Croft, Shayna Sharma, Nihar Godbole, Gregory E. Rice, Carlos Salomon

**Affiliations:** 1Exosome Biology Laboratory, Centre for Clinical Diagnostics, University of Queensland Centre for Clinical Research, Royal Brisbane and Women’s Hospital, The University of Queensland, Herston, QLD 4029, Australia; p.kalita@uq.edu.au (P.K.-d.C); shayna.sharma@uq.net.au (S.S); n.godbole@uq.net.au (N.G); g.rice@uq.edu.au (G.E.R); 2Faculty of Medicine, University of Queensland Centre for Clinical Research, Royal Brisbane and Women’s Hospital, The University of Queensland, Herston, QLD 4029, Australia

**Keywords:** ovarian cancer, liquid biopsy, sEVs, tumor microenvironment

## Abstract

Ovarian cancer (OC) is one of the most diagnosed gynecological cancers in women. Due to the lack of effective early stage screening, women are more often diagnosed at an advanced stage; therefore, it is associated with poor patient outcomes. There are a lack of tools to identify patients at the highest risk of developing this cancer. Moreover, early detection strategies, therapeutic approaches, and real-time monitoring of responses to treatment to improve survival and quality of life are also inadequate. Tumor development and progression are dependent upon cell-to-cell communication, allowing cancer cells to re-program cells not only within the surrounding tumor microenvironment, but also at distant sites. Recent studies established that extracellular vesicles (EVs) mediate bi-directional communication between normal and cancerous cells. EVs are highly stable membrane vesicles that are released from a wide range of cells, including healthy and cancer cells. They contain tissue-specific signaling molecules (e.g., proteins and miRNA) and, once released, regulate target cell phenotypes, inducing a pro-tumorigenic and immunosuppressive phenotype to contribute to tumor growth and metastasis as well as proximal and distal cell function. Thus, EVs are a “fingerprint” of their cell of origin and reflect the metabolic status. Additionally, via the capacity to evade the immune system and remain stable over long periods in circulation, EVs can be potent therapeutic agents. This review examines the potential role of EVs in the different aspects of the tumor microenvironment in OC, as well as their application in diagnosis, delivery of therapeutic agents, and disease monitoring.

## 1. Introduction

Ovarian cancer is one of the most common gynecological cancers with a fatal outcome when detected at an advanced stage, and this is partly due to the unambiguous nature of the clinical symptoms. Globally, more than 295,000 women are diagnosed with this cancer and 184,000 succumb to this disease every year [1]. The five-year survival rate of high-grade OCs is 30%, with most deaths occurring within two years of diagnosis. According to the Federation of Gynecology (FIGO) staging, stages IIIc and IV are the most commonly diagnosed stages (75%) in OCs [2]. OCs can be classified into more than 10 distinct histological subtypes, and malignant epithelial ovarian carcinomas (EOCs) comprise about 90% of the cases; therefore, this review focuses on EOCs. In 2014, the World Health Organization (WHO) classified EOCs into five main subtypes: (a) High-grade Serous Carcinoma (HGSOC); (b) Low-grade Serous Carcinoma (LGSOC); (c) Endometroid Carcinoma (EC); (d) Clear Cell Carcinoma (CCC); and (e) Mucinous Carcinoma (MC) [3]. Molecular characteristics associated with OC histologies typically range from mutations in *TP53* to *KRAS*. Briefly, HGSOC is characterized by mutations in *TP53,* LGSOC in *KRAS* and *BRAF* [4], CCC in *ARID1*α*, PIK3CA* mutations and amplifications, EC in *ARID1*α*,*
*β catenin* and *PIK3CA* mutations, and *PTEN* loss of homozygosity [5,6,7,8,9,10].

Morphological and molecular profiles categorize epithelial ovarian cancers into two main types [11]. Type I tumors are characterized by genetically stable mutations in genes, such as *PTEN, BRAF, KRAS, CTNNB1* and *PIK3CA*. Low-grade tumors, such as low-grade serous, low-grade endometrioid, clear cell and mucinous carcinomas, are Type I tumors. Type II tumors are rapidly growing, highly aggressive neoplasms that lack well-defined precursor lesions. These include high-grade serous carcinoma, malignant mixed mesodermal tumors (carcinosarcomas), and undifferentiated carcinomas. Type II tumors are characterized by mutations of *TP53* and *CCNE1*, and a high level of genetic instability. The cellular origin of HGSOCs remains to be unequivocally resolved; both the ovarian surface epithelium and fallopian tube epithelium have been reported to give rise to HGSOC [12,13,14,15].

Pelvic ultrasonography and serum CA125 concentrations are used to diagnose OCs, and a risk malignancy index is used to assess the risk of malignancy in the clinic. The risk malignancy index is a combination score of serum CA125 concentration, menopausal status and ultrasound findings. The risk malignancy index is reported to have a sensitivity of 97% [16,17]. Although there are a lack of effective screening strategies for the early detection of OCs, women with germline mutations in *BRCA1* or *BRCA2* (encoding for the DNA repair pathway proteins) are known to be highly predisposed to developing OCs. Interventions are available to reduce the risk for these patients in the form of elective surgeries to remove the ovaries and fallopian tubes. Nonetheless, overall current screening modalities have not demonstrated definitive improvements in patient mortality [18,19].

Cytoreductive surgery and platinum-based chemotherapy, including cisplatin or carboplatin and taxanes (paclitaxel or docetaxel) in combinatorial or neoadjuvant formats, remain first line treatments. Despite recent advances, interventions become ineffective over time, and 80–85% of patients develop chemoresistance [20].

The exchange of molecular signals is a crucial characteristic in cell invasion and metastasis [21,22,23]. Extracellular vesicles have been reported to play a significant role in cell-to-cell communication and have been implicated in tumor formation and metastatic disease. For example, pharmacological suppression of EV uptake at metastatic sites [24], as well as reducing the release of EV from tumor cells [23,25], reduces the formation of pre-metastatic niches and metastases. A review of the available evidence implicating EVs the development and progression of ovarian cancer is, therefore, both warranted and timely to inform further research and development.

### 1.1. Biogenesis and Release

Extracellular vesicles can be subdivided into three main groups: sEVs (50–150 nm), microvesicles (50–1000 nm), and apoptotic vesicles (>1000 nm) [26]. These vesicles differ not only in size, but also in their site of origin and biogenesis. Small EVs have an endosomal origin and their biogenesis begins with an inward budding of the plasma membrane [27]. This leads to the formation of an endosome that eventually becomes a multivesicular body through inward invaginations of its membrane. The multivesicular endosomal body contains intraluminal vesicles, and these multivesicular bodies can either fuse with the plasma membrane and release sEVs into the intercellular space or fuse with lysosomes leading to degradation [28,29]. Evidence suggests that the packaging of biomolecules into sEVs depends on the physiological state of the cell [27]. Overall, the biogenesis of sEVs is complex and context-dependent, including the cell type, stimuli, and other signals. Nonetheless, sEVs comprise various intra-vesicular components, as discussed below.

### 1.2. Intra-Vesicle Compartmentalization

Small EVs are enriched in a variety of biomolecules, ranging from nucleic acids (RNA, DNA, miRNA and non-coding RNA) to lipids and proteins [26]. Originating from endosomes, sEVs are enriched in intra-vesicular proteins, such as tetraspanins, Tsg101 and Alix. Furthermore, they may be enriched in membrane-fusion proteins, such as annexins, Rab, and flotillins and major histocompatibility complex proteins (MHC I and II). The lipid contents range from phosphatidylserine to cholesterol, and most of these are essential for maintaining sEV morphology and biogenesis. Small RNAs and non-coding RNAs have distinctive roles in recipient cells; in addition, they differ according to the cell of origin and pathophysiological states [30,31]. The enormous heterogeneity of the sEV content led to recent work by Jepessen and colleagues, who performed high-resolution gradient fractionation more comprehensively to define exosomal contents using proteomics and RNA profiling [32]. Distinctive proteomic and RNA profiles were identified between vesicular and non-vesicular compartments. This study challenged the current dogma where most diverse molecules are thought to be part of the tetraspanin-enriched sEVs. International adoption of standardized workflows and analytical procedures is requisite to unequivocally elucidate the role of exosomal signaling in human health and disease. With respect to cancer, this review primarily focusses on the role of sEVs in ovarian cancer.

## 2. Roles of Extracellular Vesicles in the Microenvironment of Ovarian Cancers

Tumor growth and survival depends upon microenvironmental conditions that promote metabolic reprogramming, angiogenesis and cellular invasion [33,34]. The extracellular matrix (ECM) and stromal cell types participate in the development of an appropriate microenvironmental niche. The ECM contains chemokines, cytokines, matrix metalloproteinases (MMPs) and cell types including cancer-associated fibroblasts, endothelial cells, and immune cells [33]. In ovarian cancer, exosomal signaling between cells regulates angiogenesis and the reprogramming of the microenvironment to promote tumorigenesis and pre-metastatic niche formation [35,36]. Indeed, the sharing of oncogenic molecules between the tumor cells and its neighbors can be achieved by the transfer of sEVs. Specifically, the ovarian cancer microenvironment is modulated through sEVs during dissemination, and interaction with stromal cells and the immune system (Figure 1). We discuss the different components of the TME and how sEVs synchronize this interplay of the cells and the ECM to promote tumor growth and survival.

### 2.1. Ovarian Cancer Dissemination

The process of ovarian cancer metastasis remains to be fully elucidated, although known pathways for metastatic spread include the lymphatic system [37], hematogenous transport [38], and via the peritoneal cavity [39]. The shedding of cancer cells from the primary site to the peritoneal cavity is one of the first steps in the dissemination of OCs. The cancer cells may attach to the surface of peritoneal organs, such as the omentum, which is the underlying stroma of the peritoneal cavity, covered by a lining of mesothelial cells [40]. The formation of a premetastatic niche is facilitated by sEVs in various cancers, including ovarian cancer [41]. The peritoneal cavity invasion through ascites is one of the initial steps in the progression of OCs. High concentrations of sEVs are found in ascites from OC patients, and the presence of ascites in OCs is associated with poor survival [42]. Graves et al. reported that sEVs derived from the ascites of OC patients induce an invasive phenotype in epithelial OC cells. This was demonstrated via the expression of extracellular-matrix-degrading proteinases, MMP-2 and 9, and urokinase-type plasminogen activators, within the sEVs (Figure 1A) [43]. It was also demonstrated that sEVs derived from ascites of OCs have potential as diagnostics, because they express tumor-intrinsic biomarkers, such as CD24 and EpCAM (Figure 1A) [44]. RNA sequencing of ascites-derived sEVs revealed miR-6780b-5p to be facilitating the epithelial to mesenchymal transition of OC cells, in vitro and in vivo (Figure 1A) [45]. Exosomal promotion of peritoneal dissemination in the microenvironment has also been observed by Nakamura and colleagues. When OC-derived sEVs were fluorescently tagged and tracked upon co-culturing with peritoneal mesothelial cells, they transferred the cell surface glycoprotein, CD44, and induced a mesenchymal morphology in these cells. They also demonstrated that those cells acquired an invasive phenotype [46]. Similarly, exosome-derived miR-99a-5p has been shown to promote peritoneal mesothelial cell invasion by increasing fibronectin expression (Figure 1A) [47]. Furthermore, matrix metalloproteinase-1 (MMP-1) mRNA carrying EVs are enriched in ascites from OC patients. MMP1-induced apoptotic cell death and destruction of the peritoneal mesothelium barrier is now a known mechanism of EV-induced peritoneal dissemination (Figure 1A) [48].

In ovarian cancer, the shedding of cancer cells and their dissemination through the peritoneal cavity is highly impacted by sEVs, and evidence suggests that they play a central role in the first step of OC dissemination. These findings suggest that the inhibition of exosome release earlier in the cancer progression stage reduces metastatic burden. Perhaps restraining the release of sEVs from the shedding cancer cells can inhibit the crosstalk and transfer of information, leading to a reduction in tumor cancer spread. Another aspect of the ovarian cancer microenvironment is the crosstalk occurring between different stromal cells, which is discussed in detail below.

### 2.2. Stromal Cell Intercommunication

The tumor microenvironment is conditioned by various stromal cells, including cancer-associated fibroblasts (CAFs), endothelial cells, adipocytes, and inflammatory cells. Small EVs released from CAFs contain higher amounts of TGFβ1, an epithelial-to mesenchymal transition (EMT) inducer, compared to normal fibroblasts. When ovarian cancer cells were exposed to these sEVs, EMT was induced (Figure 1B) [49]. Furthermore, the reprogramming of normal fibroblasts, the most abundant cell type in the stroma to CAFs, is a well-known phenomenon [50]. When normal fibroblasts are treated with ovarian-cancer-cell-derived sEVs, they display a more CAF-like phenotype [51]. Recently, Lee et al. compared ovarian cancer cell sEVs that were constitutively released or induced by EDTA chelation. They performed functional characterization of the effect of these sEVs on CAFs. Investigation of biophysical properties revealed that cellular adhesion was markedly increased in CAFs treated with chelation-induced sEVs, as well as increased cell spreading. The molecular profile of sEV miRNAs was vastly different between constitutively released and chelation-induced sEVs [52]. This study highlights that sEVs released under different conditions impact the phenotype of CAFs in the microenvironment. CAFs have been shown to impart chemotherapy resistance in ovarian cancer. This interaction is mediated by sEVs released from CAFs carrying miR-21 which, when transferred into cancer cells, caused resistance to paclitaxel through reducing the expression of the apoptosis regulatory protein APAF1 (Figure 1B) [53]. These studies highlight how interchange of exosome cargo between CAFs, and the neighboring cells impacts ovarian cancer progression and response to therapy. One of the important aspects of the ovarian cancer microenvironment is the presence of various immune cells and their interaction with the tumor cells. Small EVs have been shown to transfer information from tumor cells to help them evade the immune system.

### 2.3. Immune System Evasion

Release of sEVs from immune cells (e.g., by B-lymphocytes [54] and dendritic cells [55]) is well documented. Evasion from immune surveillance is a principal mechanism by which tumors develop within microenvironments. SEVs function as carriers of various immunomodulatory molecules for pro-tumorigenic activities. Exosomal cargo, such as miRNA 222-3p derived from OCs, has been demonstrated to induce the polarization of M1 macrophages to the more pro-tumorigenic M2 phenotype, which is associated with inducing the enhanced proliferation and invasion of cancer cells (Figure 1C) [56]. Moreover, the STAT3 pathway was found to be highly upregulated in the TAMs [56], and was associated with the immunosuppressive activity of these cells [57]. SEVs derived from ascites of OC patients induce T cell suppression in vitro, via the Janus kinase signaling pathway, and they also express FasL (Figure 1D) [58]. T cell apoptosis and cancer metastasis was promoted by FasL expressing OC-derived sEVs through the upregulation of lysophosphatidic acid, a known T cell inhibitor (Figure 1D) [59]. Ascites-derived sEVs enhance the release of interleukin-6 (IL-6) from monocytic precursor cells, activating NF-κB and STAT3 signaling and permitting a pronounced immune evasive microenvironment (Figure 1D) [60]. 

The suppression of T cells is also mediated by sEVs carrying arginase-1 (ARG-1) (Figure 1D). These sEVs are highly abundant in the ascites and plasma of OC patients. Moreover, these sEVs travel through the draining lymph nodes and are taken up by dendritic cells to inhibit antigen-specific T cell proliferation [61]. Ascites-derived sEVs are internalized by natural killer (NK) cells and induce immunosuppression [62]. Furthermore, OC- and ascites-derived sEVs impair NK2D-mediated cytotoxicity in NK cells (Figure 1D) [63]. Phosphatidylserine present on the outer surface of sEVs induces T cell arrest, and hence creates an immunosuppressive environment (Figure 1D) [64]. Furthermore, both dendritic cell and peripheral blood mononuclear cell apoptosis have been shown to be induced by ascites-derived sEVs carrying FasL and tumor necrosis factor-related apoptosis-inducing ligands (TRAIL) (Figure 1D) [65]. Shenoy et al. reported that tumor-associated sEVs suppress T cell function through various mechanisms. This was demonstrated during T cell receptor-dependent activation in a few different ways, including the translocation of NF-κB and NFAT into the nucleus, the upregulation of CD69 and CD107a, enhancing cancer cell proliferation, and producing cytokines that cause T cell suppression (Figure 1D). Moreover, stimulation and activation of CD8+ T cells by cognate viral peptides was suppressed by the sEVs. It was also reported that this change was simultaneous with binding and internalization of the sEVs [66]. Overall, various reports in the literature indicate a multifarious role of sEVs in the induction of an immunosuppressive microenvironment in OCs. This also suggests that our increasing knowledge of these mechanisms may aid in the design therapies to overcome this immunosuppression. Within the context of the ovarian cancer microenvironment and the role of sEVs in imparting a tumor-conducive environment, many model systems are being employed.

**Figure 1 cells-10-02272-f001:**
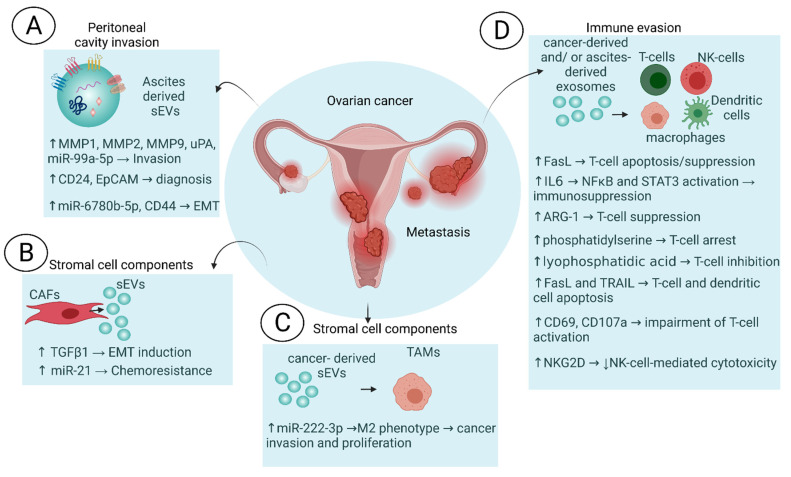
The complex ovarian cancer microenvironment and the role of EVs. EVs may be involved in multiple processes that regulate the tumor microenvironment. Metastasis is promoted through the delivery of EVs to target cells; EVs released from cancer-associated fibroblasts can lead to reprogramming of the ECM as well as chemoresistance; EVs released from cancer cells may be carrying immunosuppressive cytokines. (**A**) Peritoneal cavity invasion [41,42,43,44,45,46,47,48]; (**B**) Cancer-associated fibroblasts [49,53]; (**C**) Tumor-associated macrophages (TAMs) [55]; (**D**) Immune evasion [56,57,58,59,60,63,64,65]. *Created with BioRender.com.*

## 3. Studies of Extracellular Vesicles in Ovarian Cancer Using Various Model Systems

Various physiologically relevant model systems are used to study the role of EVs in ovarian cancer. These include 2D monolayers, animal models, and 3D models. Studies indicate that the biogenesis, packaging, and release of EVs is highly dependent and sensitive to the cellular microenvironment and on the in vitro and in vivo modelling systems utilized [67,68]. Currently, the isolation of EVs from cell condition media (CCM) predominantly relies on the uses of conventional 2D-cultured cells obtained from the tissue culture plastics [35,69,70]. Cells cultured in 2D monolayers, however, exhibit apical–basal polarity and characteristic gene expression and miRNA splicing [71]. It is highly plausible that the concentrations and/or contents of EVs released from cell grown in 2D cultures differ from those in 3D culture systems.

### 3.1. Two-Dimensional (2D) Models in Ovarian Cancer

Over the past few decades, cancer research has relied on two-dimensional cell culture to unravel the complexity of tumors. Two-dimensional monolayers are quick and cost-effective and have been invaluable in elucidating numerous mechanisms involved in tumorigenesis and used in in vitro drug discovery. The environmental conditions and nutrient supply of 2D cultures can be easily regulated; there is extensive literature, which enables comparisons of outcome measures. Furthermore, it is believed that immortalized cancer cell lines may consistently display homogenous phenotypes over multiple passages, which could be a golden standard in preclinical studies [72].

To date, studies using 2D cell culture models have identified factors involved in ovarian cancer development, metastasis, and chemotherapy resistance. Of note, Metastasis-Associated Lung Adenocarcinoma Transcript 1 (MALAT1) lnRNA has been identified as an exosomal lnRNA that was found to be released from ovarian cancer cells and imparted proangiogenic properties. In addition, elevated serum concentrations of MALAT1 were associated with metastatic disease and poor overall survival [73]. Similarly, utilizing in vitro and in vivo models, Alharbi and colleagues demonstrated that sEVs derived from highly invasive cell line (SKOV3) enhanced metastasis in vivo compared to a low-invasive cell line (OVCAR-3) [35]. Evidence suggests that sEVs released from certain ovarian cancer cells can induce EMT in other cell types. For instance, HEK293 cells were found to take up sEVs released from two different ovarian cancer cell lines, OV420 and IGROV1. Small EVs from IGROV1, however, were found to induce EMT pertaining to the exclusive elevated expression of the RNA-binding protein LIN28a [74]. Overall, 2D models have incrementally increased our knowledge about ovarian cancer. The readily available assays such as proliferation, migration and invasion can be easily performed in 2D monolayers, and some of these can be quite cost-effective to run [75].

Although of utility in elucidating mechanisms involved in the development of ovarian cancer and therapeutics, 2D models have shortcomings. In 2013, a genomic comparison between cell lines and primary samples highlighted the importance of selecting cell line models for studying ovarian cancer because many cell lines may not completely recapitulate the primary disease subtype [76]. A key feature of metastatic cells is the capacity to respond to stimuli generated by neighboring cells and ECM components [77]. Two-dimensional monolayer systems (Figure 2), however, display limited capacity to mimic ECM interactions and matrix degradations [77]. They also fail to mimic the physiological gradients of nutrients, diffusion of gases, and waste products. Cells cultured in 2D monolayers do not possess the same architecture as in vivo; therefore, it is difficult to predict the therapeutic response based on 2D experiments. Although multiple attempts have been made to mimic the physiological microenvironment in 2D model systems, wherein the plastic surfaces of the tissue culture flasks are coated with the ECM components and cells are allowed to grow in serum-free conditions [78], these efforts have never been completely successful in mimicking the pathophysiological conditions of complex 3D microenvironments [78].

### 3.2. In Vivo Model System in Ovarian Cancer

With regard to recapitulating the pathophysiology of ovarian cancers, mouse models have been the most conventional in vivo models employed to study ovarian cancer, due to their similarity to the human genome. Designing a physiologically relevant mouse model depends on the following factors: (1) the source of induced tumor cells; (2) the location of transplanted tumor cells; and (3) immune status of the mouse [79]. Hence, multiple mouse models have been designed in recent years—orthotopic models, patient-derived xenograft (PDX) models, humanized mouse models, genetically engineered mouse models and syngeneic models. To date, however, the only model closely emulating HGSOC is PDX. Due to its multiple benefits as an in vivo model, the PDX cell line has been a promising avenue in anticancer treatment and drug discovery [80,81,82,83].

Utilizing genetically engineered mouse models, Iyer and colleagues demonstrated how these can be used to advance the study of ovarian cancer immunotherapy responses, because the tumors formed in these mice closely recapitulate human disease and they also identified new mechanisms of immune-checkpoint inhibitor resistance [83]. Similarly, when the capability of a humanized mouse model of ovarian cancer to form tumors was tested using dissociated ovarian cancer biopsies injected into NSG mice, widespread metastasis was evident along with micrometastases [84]. Studies involving the role of tumor EVs in vivo can be categorized into two different types. The first category of the experiments involves the injection of genetically modified tumor cells tuned for EV secretion into animal models. It can be used to study EV secretion from the tumor cells and track the tumor-forming and metastatic capacity of these cells [23,25,85]. With respect to ovarian cancer, Dorayappan et al. demonstrated this elegantly by injecting ovarian cancer cells co-cultured with hypoxia-associated exosomes (HeX) into immunocompromised mice. Compared to normoxia-associated exosome-exposed cells, HeX induction significantly increased tumor metastatic burden. Furthermore, they observed increased serum concentrations of exosomes along with protein expressions of pSTAT3 and MMP2/9. They also exposed fallopian tube cells to HeX and injected them into the ovarian bursa and found that the fallopian tubes showed hyperplasia [86]. This study is a great example of how using in vivo models of ovarian cancer, the role of sEVs was elucidated in tumor growth and metastasis. One of the drawbacks of this approach is that the genetic modifications affecting the secretion of EVs can also affect the development and progression of cancer [23,25,87,88]; the second approach is the subsequent injection of EVs isolated from tumor cells directly into animal models. In these sets of experiments, the EVs isolated from tumor cells can be injected intraperitoneally or directly into the peripheral circulation, and subsequently taken up by stromal cells locally or at a distant site [35,89]. Subsequent events, therefore, can lead to local alterations in the TME or the formation of pre-metastatic niches at distant sites [90]. Studies further suggested that these EVs can alter the immune response or the phenotype of stromal cells [91,92]. In ovarian cancer, when we injected sEVs derived from a highly invasive cell line compared to low-invasive capacity cell line, the highly invasive cell line induced extensive metastatic niches. Proteomics analysis on these metastatic tumors revealed these changes to be associated with the Wnt canonical pathway [35]. These studies demonstrate how in vivo models assist in illuminating the role of sEVs in ovarian cancer.

Even though, in contrast to cell cultures, experiments conducted in vivo are more physiologically relevant and better mimic tissue complexity, there are some limitations. For example, animal models used in the in vivo studies are usually immunocompromised, and thus lack a critical component of the tumor microenvironment—the immune system, which plays a major role in anti-tumor and drug response modelling. Additionally, it is not yet clear that murine TME precisely emulates the human system. Such differences may compromise pre-clinical studies involving novel therapeutics. Furthermore, these concepts are generally applicable for all cancers, including ovarian cancer. There are high failure rates in human clinical trials even after obtaining promising results from in vivo mice experiments. To date, only 5% of anticancer therapeutics evaluated in mice models have been clinically licensed for human use, thus pointing to the necessity of an improved modelling system [93,94]. One issue of utmost important in ovarian cancer is the various histological subtypes which are molecularly distinct from each other. Therefore, this calls for a more personalized approach, which may be achievable through a three-dimensional model.

### 3.3. Three-Dimensional (3D) Models in Ovarian Cancer

Developing a three dimensional (3D) organotypic model that better mimics the tissue complexity and circumvents the convolutions involved in in vivo experiments may have significant utility in resolving the limitations of current in vivo and in vitro models [95]. An ideal 3D cell culture model should better stimulate the pathological and pathophysiological microenvironments, wherein the cells cultured should proliferate and differentiate. The model should be favorable for cell–cell and cell–extracellular matrix (ECM) interactions, distributed oxygen tension, sufficient nutrient, metabolic waste gradient, and tissue-specific stiffness. The 3D culture technique can broadly be classified into different types: (a) non-scaffold-based (anchorage-independent); (b) scaffold-based (anchorage-dependent); and (c) hybrid 3D cultures [96]. As a comparison, we have summarized the various advantages and disadvantages of in vitro, in vivo and 3D models in ovarian cancer research (Figure 2).

To date, multiple studies have focused on unravelling the complexity of the TME using 3D culture models to closely recapitulate the in vivo environment and conditions amenable for changes in gases and nutrient gradient, and cell-cell and cell-ECM interactions [97,98,99,100,101].

The application of 3D culture models for ovarian cancer research is relatively new; therefore, it is germane to review recent seminal contributions, such as studies by Levanon et al. and Lawrenson et al., which have contributed to unravelling mechanisms involved in the carcinogenesis of ovarian tumors from the fallopian tube. In their ex vivo model, Levanon et al. observed a high fidelity of the culture in recapitulating the in vivo cellular environment. The model was also used to analyze the role of fallopian tube secretory epithelial cells (FTSEC) in DNA repair when subjected to genotoxic stress [102]. Due to the small-scale nature and short-term viability of ex vivo culture, in vitro models were preferred [102].

In 2013, Lawrenson et al. employed an in vitro spheroid-based culture model developed from primary patient-derived FTSECs, to compare the variation in more than 1000 genes with that observed using monolayer models. Interestingly, they found that the genomic profiles for the 3D model were identical to normal FTSECs in vivo and more consistent compared to 2D monolayers, eventually indicating better mimicry of FTSECs in vivo [103,104]. Other studies [104,105,106,107,108] have focused on the role of the extracellular matrix in HGSOC metastasis. Barbolina et al. developed a 3D model using DOV13 ovarian cancer cell lines cultured on a 3D collagen Type I matrix. This study indicated the majority of fibroblasts and ECM in the progression and metastasis of ovarian cancer [105,106]. Another study, by Kenny et al. (2007), used a 3D organotypic culture model to elucidate the role of omental fibroblasts in the adhesion, invasion and proliferation of primary ovarian cancer cells during metastasis to the mesothelium [107]. An in vitro synthetic hydrogel-based model was used for embedding ovarian cancer cells by Loessner et al., who reported that cell-integrin adhesion and proteolytic remodeling regulates the proliferation of ovarian cancer cells. When compared to 2D monolayer cultures, however, Loessner observed higher survival rates in 3D spheroids when treated with paclitaxel [108,109].

With respect to EVs obtained from 3D cultures, data obtained from other cancers indicate that these EVs may be more representative of patient-derived EVs [110]. Organoids derived from a variety of cells release heterogenous EVs that accurately resemble the human physiology [111]. In cervical cancer, a 96% similarity between the 3D culture-derived EV miRNAs and patient-derived EV miRNAs was observed [67]. Furthermore, this study also identified several microRNAs common to both 3D culture-derived EVs and EVs from patient plasma [67].

Overall, 3D culture models were able to not only closely mimic the tissue complexity and TME in ovarian cancer, but also were representative models for the development, progression, invasion, and metastasis of the disease. Three-dimensional models, nevertheless, are not without limitations. The models lack method standardization and reproducibility and have a lower throughput when compared to 2D models. Additional studies are required, which may compare all three models simultaneously to understand how each of these may help us decipher the problems associated with ovarian cancer. Furthermore, a comparative study on the sEVs secreted from each of these models and the analysis of the content of the sEVs may aid in understanding how close the sEV contents are to each other. This will help in selecting the most relevant model for understanding and developing therapeutics for ovarian cancer.

## 4. Clinical Applications and Outlook

Global incidence and mortality from ovarian cancer is increasing each year, and there is a desperate need for advanced technologies to diagnose it earlier. Thus, sEVs may have various clinical applications from biomarkers of earlier detection to therapeutic interventions and the real-time monitoring of ovarian cancer (Figure 3). A list of biomarkers and their sources with cited references for ovarian cancer is presented in Table 1. The use of exosome-associated proteins, lipids and RNAs as biomarkers for ovarian cancer have several advantages, including their resistance to degradation, cell of origin specificity, potential for enrichment, and the ease of simultaneous isolation of different types of biomarkers. sEVs released from cancer cells not only contain miRNA biomarkers, but also circular [112,113,114] and long non-coding [115,116,117] RNA. Although it remains to be unequivocally established that the exosomal RNA copy number is sufficient to alter target cell phenotypes [118], exosomal RNA complements do appear fit-for-purpose as clinically useful biomarkers. Here, we specifically discuss the applications of sEVs in earlier detection as well as in the therapeutics of ovarian cancer.

### 4.1. Extracellular Vesicle Biomarkers for the Earlier Detection of Ovarian Cancer

The goal of population-based screening and earlier detection of OC is to identify cancerous cells while they are still confined within the ovary (i.e., Stage 1) or before migration from their primary site of origin, thus allowing intervention that improves patient outcomes. Small EVs carrying specific signatures of cancerous growth can be of clinical benefit as they circulate in the system and can be easily detected through a non-invasive blood test. Detecting early biomarkers in sEVs may help in deciding if oophorectomy or salpingectomy may be of clinical benefit. Consequently, bilateral salpingo-oophorectomy has been proposed as a risk-reduction strategy in high-risk patients with BRCA mutations [119]. The diagnosis of early stage ovarian cancer may be further confounded by the presence of slow (Type I) and rapid (Type II) growing tumors and the possibility that conventional biomarker discovery approaches which target Stage 1 ovarian cancer may be biased towards the detection of Type I cancers and miss the more common and aggressive Type II cancers [120]. A more productive and clinically relevant strategy may be to target stage-independent molecular biomarkers of low-volume Type II carcinomas. Such biomarkers, either individually or incorporated into multivariate index assays (MIAs), may aid in the earlier identification of OCs with the highest prevalence and mortality rate, and afford a realistic opportunity to improve clinical outcomes.

Previous approaches to develop population-based screening and earlier detection tests for OC have failed to deliver a specificity that is clinically acceptable [19,121,122,123,124,125,126,127]. A specificity of 99.6% or more is widely accepted as requisite for an OC population-based screening test. Such a test would result in no more than nine women undergoing unnecessary procedures consequent to false-positive results. It is axiomatic that in at-risk subpopulations, where the prevalence of ovarian cancer is greater, a test of lower specificity may be of utility.

To date, efforts to develop and validate tests to aid in the assessment of ovarian cancer and adnexal masses have almost exclusively focused on soluble biomarkers, individually (e.g., CA125, HE4, CEA, VCAM), in MIAs (e.g., FDA-approved tests, OVA1, Overa, Risk of Malignancy Algorithm), or in combination with imaging modalities (e.g., Risk of Malignancy Index). More recently, the potential for EV biomarkers to deliver more informative ovarian cancer tests has been recognized [29,35,69,128,129,130,131,132,133,134,135,136]. With regard to diagnostics, the miRNA profile of circulating EpCAM-positive sEVs is reflective of that observed in ovarian tumor biopsies (Table 1) [137]. In this case, miRNA has been packaged into sEVs that are released by ovarian cancer cells into extracellular fluids and is protected from degradation by ribonucleases. This exosomal eight miRNA (miR-141, miR-21, miR-200a, miR-200b, miR-200c, miR-214, miR-205 and miR-203) profile has also been reported to be reflective of the stage of ovarian cancer (Figure 3A) [137]. Similarly, ovarian-cancer-derived sEVs express CD24 and EpCAM, which are tumor-exclusive markers and therefore can differentiate between cancerous and non-cancerous lesions (Figure 3A) [44]. In addition, miR-200a, miR-200b, miR-200c and miR-373 are elevated in the ovarian cancer patients compared to healthy controls (Figure 3A) [138].

Interestingly, Pan and colleagues reported higher concentrations of sEVs miR-23a and miR-92a in patients with epithelial ovarian cancer when compared to patients with ovarian cystadenoma (Figure 3A) [139]. Furthermore, sEV miR-200b expression was five times greater in patients with epithelial ovarian cancer, and associated with increased ovarian cancer cell proliferation, as well as poorer overall survival. Higher sEV miR-200c expression was also noted in the serum of patients with HGSOC and non-HGSOC, compared to the non-cancer group [140]. In addition, exosomal miR-93 and miR-145 expression was also increased (Figure 3A) [140]. In contrast, miR-34a expression was increased in patients with early stage disease compared to patients with advanced stage disease, patients with lymph node metastases, and recurrence (Figure 3A) [141].

In addition to miRNAs, proteins and lipids associated with sEVs are informative of the metabolic and metastatic status of the cell of origin. A lipidomic and proteomic analysis comparing sEVs obtained from SKOV-3 cells showed greater expression of ChE and ZyE lipid species compared to sEVs from an ovarian surface cell line (HOSEPiC), as well as abundant expression of Collagen Type V Alpha 2 chain (COL5A2), and lipoprotein lipase (LPL) (Figure 3A) [132]. Differential sorting of both protein and lipids into sEVs has been observed in ovarian cancer cell lines (SKOV-3,) when compared to ovarian epithelial cell of non-malignant origin (HOSEPiC) (Figure 3A) [132]. Zhang and colleagues identified LBP (lipopolysaccharide-binding protein), GSN (gelsolin), FGA (fibrinogen alpha chain), and FGG (fibrinogen gamma chain) as potential diagnostic exosomal proteins for ovarian cancer (Figure 3A) [134], with FGA found to be the most promising diagnostic marker. These studies indicate that miRNA signatures from sEVs could be potential diagnostic biomarkers for the early detection of ovarian cancer. Apart from earlier detection, circulating sEVs have great potential as therapeutic delivery vehicles as well as treatment response monitoring, because the signatures carried in these sEVs may be different in patients who respond compared to patients who do not respond to therapy.

### 4.2. Small EVs as Therapeutic Delivery Vehicles and Therapy Response Monitors for Ovarian Cancer

EVs have the potential to be clinically useful bio-carriers for gene and drug delivery [128,142], which can be leveraged for treating ovarian cancer; however, several key factors have impeded their implementation, including batch-to batch variation due to the inadequate standardization of protocols, a lack of large-scale production capabilities, and time-consuming isolation protocols [143,144]. Nonetheless, due to their relative stability and biocompatibility, research and development into their use as delivery vehicles is expanding [145,146]. Recently, for example, mesenchymal-stem-cell-derived sEVs have been efficiently produced in scalable quantity [147]. To date, few studies have investigated the used of EVs to target ovarian cancer [148,149]. Triptolide-loaded sEVs were delivered in vitro and in vivo to test their efficacy against ovarian cancer. Although delivery and bioavailability were successful, the drug itself had toxic effects on the kidneys and the liver [148]. Encapsulation of doxorubicin in sEVs and tumor targeting using iRGD peptides on sEVs surface led to reduced tumor burden compared to free doxorubicin (Figure 3B) [150]. Exosomal doxorubicin was also found to increase its therapeutic index in HGSOC mouse models (Figure 3B) [149].

Interestingly, a recent study proposed the use of immune-derived sEVs mimetics (IDEM), another alternative to upscale production and increasing bioavailability. IDEMs have a similar profile to sEVs, and when engineered to deliver doxorubicin utilizing 2D and 3D models of ovarian cancer, they displayed higher yields and increased drug uptake and cytotoxicity compared to free doxorubicin (Figure 3B) [151]. In xenografted mice, CRISPR/Cas9 plasmid-loaded sEVs efficiently suppressed the expression of poly (ADP-ribose) polymerase-1 (PARP-1), inducing apoptosis in ovarian cancer and making the cells more sensitive to cisplatin (Figure 3B) [152].

**Figure 3 cells-10-02272-f003:**
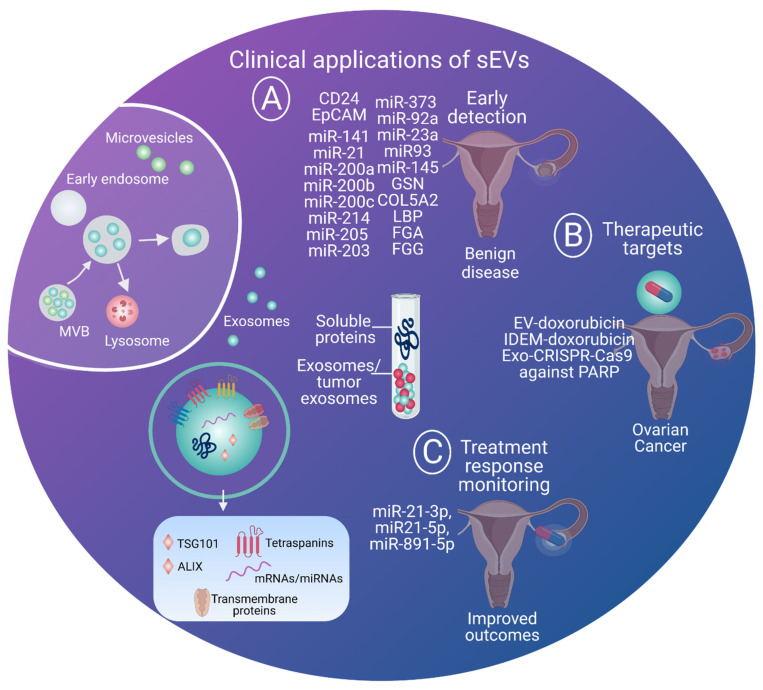
An overview of various clinical application of EVs. The biogenesis of EVs is a continual process in the body as it progresses through different stages. Through non-invasive processes such as blood collection, EVs, especially sEVs, could be excellent biomarkers. Biological fluids can carry not only soluble proteins, but also cancer sEVs, which may contain important information to track the disease. These sEVs can be versatile and may aid in early detection when the benign disease turns malignant. They can also be used as a delivery vehicle for targeted therapies due to their size. Over time, the exosomal biomarkers can be used to predict treatment response as well as monitoring. (**A**) Early detection [44,132,137,138,139,140,141]; (**B**) Therapeutic targets [149,150,151,152]; (**C**) Treatment response monitoring [69]. *Created with BioRender.com.*

Small EVs serving as potential biomarkers for predicting response to therapy have been discussed in length by Zhou and colleagues. With respect to ovarian cancer, circulating exosome concentrations may be indicative of chemotherapy responsiveness [153]. The authors reported that patients who responded to chemotherapy had increased or decreased levels of exosomal proteins after chemotherapy compared to non-responders. Exosomal cargo could potentially be used to predict chemotherapy resistance. For example, Alharbi et al. observed that miR-21-3p, miR-21-5p and miR-891-5p imparted chemotherapy resistance in ovarian cancer (Figure 3C) [69]. Development of better prognostic biomarkers for ovarian cancer will lead to better and personalized treatment plans. This will not only reduce the overtreatment of patients who might not respond to a particular therapy, but also provide better care for patient who are likely to respond to a treatment. This will ultimately have better outcomes for patients and reduce the economic and health burden on the system.

## 5. Conclusions and Perspectives

Market analysis reports predict that EV research and applications are projected to be a high growth area. Grand view research, Inc. (San Francisco, CA 94105, USA) published a market report in 2018, which estimated that exosome research and its clinical use will be worth USD 2.28 billion by 2030, representing an annual growth rate of 18.8% [154]. Taking this into account, and the recent discoveries that exosomal-signaling pathways regulate key aspects of ovarian cancer metastasis and disease progression, EVs are an attractive avenue to explore for ovarian cancer detection, therapeutics, and prognosis. Early detection of ovarian cancer remains a significant analytical challenge and an important clinical need. The development of tests for early detection would allow the impact of earlier detection on disease outcome to be assessed [155]. If proven to be effective in increasing 5-year survival rates, the implementation of community-based screening programs, similar to those currently available for breast cancer, would dramatically decrease mortality and morbidity. A more complete understanding of the mechanisms regulating the packaging of proteins, lipids, and nucleic acids into sEVs by both malignant and normal ovarian cells is a requisite to realizing these opportunities. Furthermore, elucidating the effects of ovarian-cancer-cell-derived sEVs on neighboring cells and their extracellular milieu will provide new insights into the development of therapeutic modalities, including the use of inhibitors of exosome release or repressors of exosomal RNA activity. To fully realize the potential and clinical utility of exosomal biomarkers in the earlier identification of ovarian cancer in women, and to monitor disease progression, robust and standardized translation pathways must be implemented within research laboratories, including the conduct of biomarker discovery and the development of in vitro diagnostics, in compliance with appropriate regulatory guidelines (e.g., ISO13485 and 21CFR part 809). The lack of compliance with such standards significantly contributes to the high rate of translational failure for ovarian cancer diagnostic and prognostic modalities. The specialized and labor-intensive methodologies of EV isolation also present a technical hurdle in the application of EVs in a clinical diagnostic laboratory. Therefore, newer technologies that circumvent the current practical limitations will be critical for implementation in the clinic, as will standardized protocols for isolation and analysis. Nonetheless, the future holds a promising avenue for EV applications in ovarian cancer diagnosis and monitoring.

## Figures and Tables

**Figure 2 cells-10-02272-f002:**
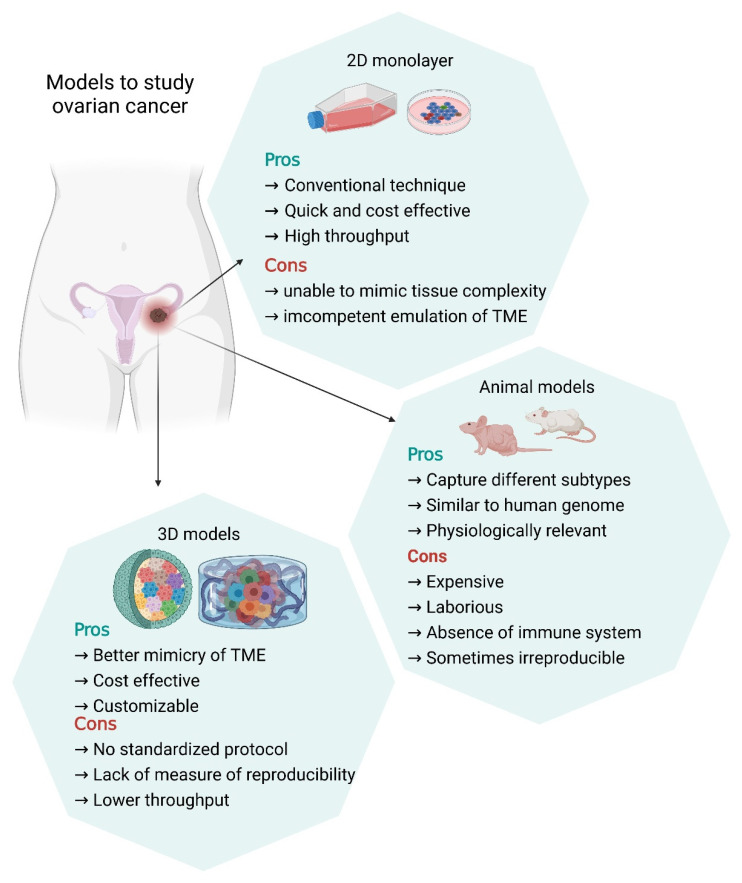
Advantages and disadvantages of the various models to study ovarian cancer. *Created with BioRender.com.*

**Table 1 cells-10-02272-t001:** Lists of EV biomarkers and their sources.

Biomarker Type	Biomarker	Sample Source(s)	References
Protein	EpCAM	Ovarian cancer patients with benign and malignant ovarian carcinomas	[137]
	EpCAM	Ovarian cancer cell lines—CaOV3, OV90, OVCA429, OVCAR3	[44]
	CD24	Ovarian cancer cell lines—CaOV3, OV90, OVCAR3, UCI101	[44]
miRNA	miR-141, miR-21, miR-200a, miR-200b, miR-200c, miR-214, miR-205 and miR-203	Serum	[137]
	miR-23a and miR-92a	Patients with EOC and ovarian cysteadenoma	[139]
	miR-200b	Plasma	[139]
	miR-200c, miR-93 and miR-145	Serum (HGSOC patients)	[140]
	miR-34a	Serum	[110]
Lipids	ChE, ZyE, Collagen Type V Alpha 2 chain (COL5A2), and lipoprotein lipase (LPL)	SKOV3 cell line	[132]
	LBP (lipopolysaccharide-binding protein), GSN (gelsolin), FGA (fibrinogen alpha chain), and FGG (fibrinogen gamma chain)	Plasma	[12]
Long non-coding RNA	MALAT1	SKOV3, HO8910, in vivo and Serum	[73]

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
