# Peer review of "Ovarian-Cancer-Associated Extracellular Vesicles: Microenvironmental Regulation and Potential Clinical Applications"

_cells, 2021, doi:10.3390/cells10092272_

Round 1

Reviewer 1 Report

This article reviewed the role of ovarian cancer (OC)-derived extracellular vesicles (EVs) in modulating the tumor microenvironment (TME) of OC and the potential application of EVs in monitoring OC progression. Some aspects e.g. Findings of exosomes in OC's TMC in Section 2 and Models for OC-EV studies in Section 3 are informative but the clinical application of OC-EVs in Section 4 is way too general to be discussed as a separate section in a review article.  

I agreed with most of the points included in the review but the introductory statement, organization of points of view, conclusion, and paragraphing of each section requires rigorous editing. I suggest the following structure for each section of this review:

  1. General introduction for the section
  2. Points of discussion
  3. Conclusive statement for the section
  4. What readers can expect in the following section

Below my specific comments for each section of the review:

Section 1: Introduction

The introduction of a review is expected to produce important background knowledge for readers to understand the details of the main themes of the review. The title of this review is "EVs: role in the microenvironment of OC and their utility in monitoring cancer progression" but there is a lack of introduction of the "microenvironment", or more commonly know as TME. There are tremendous studies done to define TME and how it contributes to tumor progression so it is necessary to introduce this before introduce the new element (EVs, exosomes) that was recently found to be part of TME. 

Section 2: Exosomes in the microenvironmental landscape of ovarian cancer

This is the most well-structured section in the review. However, lines 147-149 don't sound like the conclusion of the paragraph. Examples shown are only brain tumors. References for other cancer types? If this section is the introductory part of the following subsections, then a brief introduction for what the reader can respect for the following parts (2.1 - 2.3) can be included here. 

Section 3: Extracellular vesicles derived from various model systems to study ovarian cancer

All three models (2D, animal and 3D) should be fairly reviewed (each in one subsection) and compared then only justify why 3D should be focused on in future studies. 

Section 4: Clinical applications and outlook

This is the most poorly structured section with minimal details of EVs related information specific to OCs. Only one miRNA was mentioned while no specific protein and/or lipid was discussed as potential EV marker for OC. Lines 155-156 mentioned, "These changes could be monitored overtime in a non-invasive manner as biomarkers of progression". Why there is no follow up of this line in Section 4 which aims to explore the clinical potential of OC-EVs?

Figures and Tables

All the figures and table look nice but too general to be included in a research journal. The general information about EVs depicted in Figures 1 - 3 and Table 1 had been widely reviewed elsewhere. The unique feature of this review article is to compile the information specific to OC and summarize the details using figures and tables with citations of references. Figures and Tables shown in this review should be specific and informative summarizing related studies of OC-EV. 

Last but not least, cancers are diverse, complicated, and heterogeneous. It is important to specify which information is the findings general to most types of cancers, and which information is specific to the type of cancer reviewed in this article.   

Please refer to the attached manuscript for more comments. 

Author Response

Section 1: Introduction

The introduction of a review is expected to produce important background knowledge for readers to understand the details of the main themes of the review. The title of this review is "EVs: role in the microenvironment of OC and their utility in monitoring cancer progression" but there is a lack of introduction of the "microenvironment", or more commonly know as TME. There are tremendous studies done to define TME and how it contributes to tumor progression so it is necessary to introduce this before introduce the new element (EVs, exosomes) that was recently found to be part of TME.

We thank the reviewer for the comment, and we have provided an introduction of the TME in the appropriate section. Please see section 3 in the revised version of the manuscript.

Section 2: Exosomes in the microenvironmental landscape of ovarian cancer

This is the most well-structured section in the review. However, lines 147-149 don't sound like the conclusion of the paragraph. Examples shown are only brain tumors. References for other cancer types? If this section is the introductory part of the following subsections, then a brief introduction for what the reader can respect for the following parts (2.1 - 2.3) can be included here.

We thank the reviewer for the comment, and we have rewritten this section to fit the theme. Please see section 3 in the revised version of the manuscript

Section 3: Extracellular vesicles derived from various model systems to study ovarian cancer

All three models (2D, animal and 3D) should be fairly reviewed (each in one subsection) and compared then only justify why 3D should be focused on in future studies.

We thank the reviewer for the comment, and we have rewritten this section to fit the theme. Please see section 4 in the revised version of the manuscript.

Section 4: Clinical applications and outlook

This is the most poorly structured section with minimal details of EVs related information specific to OCs. Only one miRNA was mentioned while no specific protein and/or lipid was discussed as potential EV marker for OC. Lines 155-156 mentioned, "These changes could be monitored overtime in a non-invasive manner as biomarkers of progression". Why there is no follow up of this line in Section 4 which aims to explore the clinical potential of OC-EVs?

We thank the reviewer for the comment, and we have rewritten this section to fit the theme. Please see section 5 in the revised version of the manuscript.

Figures and Tables

All the figures and table look nice but too general to be included in a research journal. The general information about EVs depicted in Figures 1 - 3 and Table 1 had been widely reviewed elsewhere. The unique feature of this review article is to compile the information specific to OC and summarize the details using figures and tables with citations of references. Figures and Tables shown in this review should be specific and informative summarizing related studies of OC-EV.

We thank the reviewer for the comment, and we have added information into the figures to make them more specific. Please see the figure changes throughout the revised version of the manuscript.

Reviewer 2 Report

The manuscript entitled "Extracellular vesicles: role in the microenvironment of ovarian cancer and their utility in monitoring cancer progression " by Priyakshi Kalita-de Croft et al., it will be improved if the followings are undertaken.

- First of all, the title of the article is too general and it is advised to make it more specific, if the authors are emphasized on Ovarian cancer.

- It will be interesting to make a table listing out all the components identified in EV, e.g., RNA, proteins, or metabolites, of Ovarian cancer.

- Many of the figures are showing some general knowledge of cancer and EVs and not very specific to Ovarian cancer, which should be improved.

- The authors should make a table summarizing various studies on EVs of ovarian cancer, also listing if there is difference of EV components among various women of different ethnicity. Now is difficult to comprehend the studies mentioned.

- Again, the authors should list a table showing the early time points or later time point biomarkers (fingerprints) useful for predicting different stage of Ovarian cancer.

- Table 1 should be considered as a figure.

- Figures should be coherent in styles, such as font type and size.

- Typos and unfriendly mode of English usage can be found, which should be addressed.

Author Response

Reviewer 2

Response

First of all, the title of the article is too general and it is advised to make it more specific, if the authors are emphasized on Ovarian cancer.

We thank the reviewer for the comment, and we have changed the title. Please see the revised version of the manuscript.

- It will be interesting to make a table listing out all the components identified in EV, e.g., RNA, proteins, or metabolites, of Ovarian cancer.

We thank the reviewer for the comment, and we compiled a table addressing these comments.

Many of the figures are showing some general knowledge of cancer and EVs and not very specific to Ovarian cancer, which should be improved.

We thank the reviewer for the comment, and we have included more specific information it the figures. Please see the revised version of the manuscript.

The authors should make a table summarizing various studies on EVs of ovarian cancer, also listing if there is difference of EV components among various women of different ethnicity. Now is difficult to comprehend the studies mentioned.

We thank the reviewer for the comment, however, we found it difficult to compile a comprehensive table with information about ethnicity and therefore we have included one table listing different components of the EVs and identified cargo and the type of samples used. Please see the revised version of the manuscript.

Again, the authors should list a table showing the early time points or later time point biomarkers (fingerprints) useful for predicting different stage of Ovarian cancer.

We thank the reviewer for the comment, and we have included a table where we have listed out the cargo and the type of samples. Please see the revised version of the manuscript.

Table 1 should be considered as a figure.

Figures should be coherent in styles, such as font type and size.

We thank the reviewer for the comment, and we have standardized the fonts and sizes in the figures. Please see the revised version of the manuscript.

Typos and unfriendly mode of English usage can be found, which should be addressed.

We thank the reviewer for the comment, and we have changed our wordings around to make it more reader friendly throughout the document. Please see the revised version of the manuscript.

Round 2

Reviewer 1 Report

The structure and organization of the revised version are greatly improved when compared to the first submission. Sections 3 & 5 were nicely revised. Below the suggestions for the minor amendment:

Section 3

  • The title should be "Roles of exosomes in the tumor microenvironment of ovarian cancers"
  • Please clearly label the specific aspects TME in Figure 1 with (A), (B), (C) & (D) as suggested in the attached PDF. In the caption, the general description can be deleted and replaced with the description of the labels with references "Figure 1. The complex ovarian cancer microenvironment and the role of EVs: (A) Immune evasion (cited references), (B) Tumor-associated macrophages (cited references), (C) Cancer-associated fibroblasts (cite references), (D) Peritoneal cavity invasion (cite references)"
  • In the text, the discussion in detail for respective aspects (A), (B), (C), and (D) should be cited specifically with Figure 1A, Figure 1B, Figure 1C, and Figure 1D accordingly.
  • In another word, what you depicted in Figure 1 should be discussed in the text and the Figure serves to summarize the content in Section 3 and provide a quick glance at the main points specific for TME-EVs in ovarian cancers. 

Section 5

  • Please clearly label the specific clinical applications in Figure 3 with (A), (B) & (C) as suggested in the attached PDF. In the caption, the general description can be deleted and replaced with the description of the labels with references "Figure 3. An overview of various clinical applications of EVs: (A) Early detection (cite references), (B) Therapeutic targets (cite references), (C) Treatment response monitoring (cite references)"
  • In the text, the discussion in detail for respective aspects (A), (B) & (C) should be cited specifically with Figure 3A, Figure 3B, & Figure 3C accordingly.
  • In another word, what you depict in Figure 3 should be discussed in the text and the Figure serves to summarize the content in Section 5 and provide a quick glance at the main points specific for clinical application of EVs in ovarian cancers. 

However, a similar organization/structure and the fluent flow of discussion couldn't be observed in Section 4.

  • I believe what authors aimed to discuss in this section is "Studies of extracellular vesicles (exosomes?) in ovarian cancer using various model systems". Please amend the title accordingly.
  • I did mention in the previous reviewer report, the unique feature and focus of this review should be OVARIAN CANCERS related. Please find my comment and questions in the attached PDF.
  • I don't see an improvement for Figure 4. I strongly suggest replacing Figure 4 with a Table (or a better diagram like Figure 3 or Figure 5) that could clearly summarize the content of the improved version of Section 4. 

Last but not least, please decide which one (Extracellular vesicles or Exoxomes) to use throughout the review article. A justification for this should be included in the introduction.  

Author Response

We thank the reviewers for their insightful comments and questions relating to the manuscript. We have carefully considered their comments and addressed them in detail in the revised version of the manuscript.

Reviewer 1.

The structure and organization of the revised version are greatly improved when compared to the first submission. Sections 3 & 5 were nicely revised. Below the suggestions for the minor amendment:

Section 3

  • The title should be "Roles of exosomes in the tumor microenvironment of ovarian cancers"

Reply: Thanks for the suggestion. We have modified the title accordingly.

  • Please clearly label the specific aspects TME in Figure 1 with (A), (B), (C) & (D) as suggested in the attached PDF. In the caption, the general description can be deleted and replaced with the description of the labels with references "Figure 1. The complex ovarian cancer microenvironment and the role of EVs: (A) Immune evasion (cited references), (B) Tumor-associated macrophages (cited references), (C) Cancer-associated fibroblasts (cite references), (D) Peritoneal cavity invasion (cite references)"

Reply: We have included the modifications in the revised version of the manuscript. Please see Figure 1.

  • In the text, the discussion in detail for respective aspects (A), (B), (C), and (D) should be cited specifically with Figure 1A, Figure 1B, Figure 1C, and Figure 1D accordingly.
  • In another word, what you depicted in Figure 1 should be discussed in the text and the Figure serves to summarize the content in Section 3 and provide a quick glance at the main points specific for TME-EVs in ovarian cancers. 

Reply: We agree with the reviewer. We have discussed the signalling pathways mentioned din the figure in the main text and use the figure as summary.

Section 5

  • Please clearly label the specific clinical applications in Figure 3 with (A), (B) & (C) as suggested in the attached PDF. In the caption, the general description can be deleted and replaced with the description of the labels with references "Figure 3. An overview of various clinical applications of EVs: (A) Early detection (cite references), (B) Therapeutic targets (cite references), (C) Treatment response monitoring (cite references)"

Reply: We have included the modifications in the revised version of the manuscript. Please see Figure 3.

  • In the text, the discussion in detail for respective aspects (A), (B) & (C) should be cited specifically with Figure 3A, Figure 3B, & Figure 3C accordingly.
  • In another word, what you depict in Figure 3 should be discussed in the text and the Figure serves to summarize the content in Section 5 and provide a quick glance at the main points specific for clinical application of EVs in ovarian cancers. 

Reply: We agree with the reviewer. We have discussed the signalling pathways mentioned din the figure in the main text and use the figure as summary.

However, a similar organization/structure and the fluent flow of discussion couldn't be observed in Section 4.

  • I believe what authors aimed to discuss in this section is "Studies of extracellular vesicles (exosomes?) in ovarian cancer using various model systems". Please amend the title accordingly.

Reply: Thanks for the suggestions. We have modified the title accordingly.

  • I did mention in the previous reviewer report, the unique feature and focus of this review should be OVARIAN CANCERS related. Please find my comment and questions in the attached PDF.

Reply: Thanks for the feedback. We have incorporated all your comments in the revised version of the manuscript.

  • I don't see an improvement for Figure 4. I strongly suggest replacing Figure 4 with a Table (or a better diagram like Figure 3 or Figure 5) that could clearly summarize the content of the improved version of Section 4. 

Reply: We have modified the Figure 4 in the revised version of the manuscript.

Last but not least, please decide which one (Extracellular vesicles or Exoxomes) to use throughout the review article. A justification for this should be included in the introduction.  

Reply: Yes, we agree with the reviewer. We have used the term extracellular vesicle in the manuscript.

Reviewer 2 Report

The authors have made some efforts to improve the manuscript. My last worry is that the figures are overwhelming with the cartoon of ovaries, utherus/cervical regions, yet, it is unclear what those lesions meant? By looking at these figures, seem that the manuscript is not only focusing on the ovaries but also other parts of the female reproductive organs.  It would be better to improve the figures so that people would understand the focus of this review.

Lastly, there are still typos in the manuscript, e.g. in figure 3, should be soluble proteins, not soluble protiens. The authors must carefully check for typos in both the text and figures.

Author Response

Reviewer 2

The authors have made some efforts to improve the manuscript. My last worry is that the figures are overwhelming with the cartoon of ovaries, utherus/cervical regions, yet, it is unclear what those lesions meant? By looking at these figures, seem that the manuscript is not only focusing on the ovaries but also other parts of the female reproductive organs.  It would be better to improve the figures so that people would understand the focus of this review.

Reply: Thanks for the feedback. We have included modifications to the figures to clarify ovarian cancer and metastasis sites.

Lastly, there are still typos in the manuscript, e.g. in figure 3, should be soluble proteins, not soluble protiens. The authors must carefully check for typos in both the text and figures.

Reply: We apologise for the typos. We have carefully reviewed the updated version of the manuscript.

Round 3

Reviewer 1 Report

No comments

Author Response

We thank the reviewer for their insightful comments and questions relating to the manuscript.

Reviewer 2 Report

I am satisfied with the efforts of the authors in responding to my concerns.

Author Response

(The authors gave the same response as above.)
